# Extracellular Vesicles Allow Epigenetic Mechanotransduction between Chondrocytes and Osteoblasts

**DOI:** 10.3390/ijms222413282

**Published:** 2021-12-10

**Authors:** Xiaobin Shang, Kai Oliver Böker, Shahed Taheri, Wolfgang Lehmann, Arndt F. Schilling

**Affiliations:** Department of Trauma Surgery, Orthopedics and Plastic Surgery, University Medical Center Göttingen, 37075 Göttingen, Germany; xiaobin.shang@stud.uni-goettingen.de (X.S.); shahed.taheri@med.uni-goettingen.de (S.T.); Wolfgang.Lehmann@med.uni-goettingen.de (W.L.); Arndt.schilling@med.uni-goettingen.de (A.F.S.)

**Keywords:** osteoarthritis, microRNA, extracellular vesicles, cell–cell communication, mechanical loading

## Abstract

MicroRNAs (miRNAs) can be transported in extracellular vesicles (EVs) and are qualified as possible messengers for cell–cell communication. In the context of osteoarthritis (OA), miR-221-3p has been shown to have a mechanosensitive and a paracrine function inside cartilage. However, the question remains if EVs with miR-221-3p can act as molecular mechanotransducers between cells of different tissues. Here, we studied the effect of EV-mediated transport in the communication between chondrocytes and osteoblasts in vitro in a rat model. In silico analysis (Targetscan, miRWalk, miRDB) revealed putative targets of miRNA-221-3p (CDKN1B/p27, TIMP-3, Tcf7l2/TCF4, ARNT). Indeed, transfection of miRNA-221-3p in chondrocytes and osteoblasts resulted in regulation of these targets. Coculture experiments of transfected chondrocytes with untransfected osteoblasts not only showed regulation of these target genes in osteoblasts but also inhibition of their bone formation capacity. Direct treatment with chondrocyte-derived EVs validated that chondrocyte-produced extracellular miR-221-3p was responsible for this effect. Altogether, our study provides a novel perspective on a possible communication pathway of a mechanically induced epigenetic signal through EVs. This may be important for processes at the interface of bone and cartilage, such as OA development, physiologic joint homeostasis, growth or fracture healing, as well as for other tissue interfaces with differing biomechanical properties.

## 1. Introduction

Osteoarthritis (OA) is a disease of the whole joint characterized by chronic inflammation, cartilage degradation, and subchondral bone remodeling, which can lead to severe pain and disability in patients [1,2,3]. It is estimated that more than 600 million people over the age of 40 years are living with symptomatic knee OA, resulting in individual suffering and substantial socioeconomic costs [4,5,6]. Current treatment of OA focuses on pain therapy and inflammation treatment and the only option for endpoint OA patients is total knee arthroplasty surgery [7]. So far, no disease-modifying treatments are available, despite decades of research. Therefore, further study on physiopathologic mechanisms of joint OA is urgent to explore pioneering therapeutic regimens.

As nano-size vesicles wrapped by a phospholipid bilayer, Extracellular Vesicles (EVs) derived from chondrocytes could promote chondrocyte proliferation and migration in a paracrine fashion [8]. MicroRNAs (miRNAs), which can be transported in EVs, have been demonstrated to participate in OA pathophysiology through post-transcriptional regulation of target mRNAs [9,10,11,12]. In particular, miR-221 has been shown to be regulated in arthritic cartilage and the expression of miR-221 is upregulated under mechanical loading [13,14,15,16]. The joint is exposed to dynamic mechanical loading [17] and mechanics play a key role in the regulation of joint function and dysfunction. This leads to adaptive changes in cartilage and subchondral bone [18]. Due to its biomechanical properties, at the same load, cartilage is much more deformed than bone [18], making mechano-modulation of chondrocytes (e.g., via autocrine and paracrine signaling) a probable mechanism for the maintenance of joint homeostasis [19]. As in the joint, the subchondral bone is in direct contact with cartilage, and microchannels between the tissues theoretically allow cartilage–bone crosstalk [20,21,22,23,24]. The question remains if EVs can also modulate cells of the adjacent tissue, thereby translating the mechanical information into a tissue-border-transcending epigenetic signal.

In this study, we investigated the role of EV-mediated transport of miRNA-221-3p in the communication between chondrocytes and osteoblasts in vitro. We hypothesized that the dynamic expression of miRNA-221-3p in cartilage chondrocytes may affect the molecular activity and the bone formation capacity of osteoblasts through EVs.

## 2. Results

### 2.1. Identification of Chondrocyte Secreted EVs

EVs were successfully isolated from the conditioned supernatant of chondrocytes by ultracentrifugation. Nanoparticle tracking analysis (NTA) demonstrated that the size distribution of EVs ranged from 50 to 200 nm (Figure 1a,b). Transmission electron microscopy (TEM) further showed a typical cup—or sphere-shaped—morphology of EVs with a bilayer membrane structure (Figure 1c). Further analysis demonstrated that transfection of miR-221-3p mimics did not affect the average size and amount of EVs compared to control miRNA transfections (Figure 1d,e). Characteristic markers such as CD81, Alix, and TSG101 proteins were positively expressed in EVs based on Western blot analysis (Figure 1f and Appendix A).

### 2.2. MiR-221-3p Expression in OA

We successfully constructed an in vitro OA model with IL-1β-treated chondrocytes. Details about the expression levels of OA-related genes can be found in Appendix A. In brief, the expression level of catabolic genes MMP-13, ADAMTS-5, and COX2 in IL-1β-induced chondrocytes were significantly increased while the anabolic gene SOX-9 was decreased (Appendix A). Further analysis showed significantly downregulated expression of miR-221-3p in IL-1β-treated chondrocytes versus control (Appendix A).

### 2.3. Mir-221-3p and Target Gene Analysis in Chondrocytes and Osteoblasts

To determine the effect of miR-221-3p on chondrocytes, chondrocytes were transfected with miR-221-3p mimic and scrambled control. As shown in Figure 2a, the miR-221-3p expression was increased by ~200 fold (*p* < 0.001) after 48 h of transfection.

Thereafter, we predicted downstream target mRNAs using the publicly available websites miRDB, miRwalk, and TargetScan (Figure 2b). Given the potential role of miR-221-3p in chondrocytes and osteoblasts homeostasis as reported in the literature, we chose four possible targets including Cyclin-dependent kinase inhibitor 1B (CDKN1B/p27), Tissue inhibitor of metalloproteinase 3 (TIMP-3), Transcription factor 7-like 2 (Tcf7l2/TCF4), and aryl hydrocarbon receptor nuclear translocator (ARNT) for further research. Indeed, all of these four genes were identified as direct targets of miR-221-3p by qRT-PCR (Figure 2c–f).

To understand the potential role of miR-221-3p in osteoblast differentiation, osteoblasts were transfected with miR-221-3p mimic or scrambled control. Osteogenic differentiation was evaluated by histology and gene expression analysis. The miR-221-3p expression was increased in treated osteoblasts by ~2725 fold (*p* < 0.001) (Figure 3a). MiR-221-3p overexpression inhibited the expression of the aforementioned targets CDKN1B/p27 (85%, *p* < 0.001), ARNT (54%, *p* < 0.001), TIMP-3 (75%, *p* < 0.001) and Tcf7l2/TCF4 (60%, *p* < 0.001) (Figure 3b–e). More importantly, compared with the scrambled control, osteogenic markers such as COL1A1, RUNX2, and OCN were inhibited by ~60% (*p* < 0.001), 74% (*p* < 0.001), and 97% (*p* < 0.001) (Figure 3f–h), suggesting an influence of miR-221-3p on osteoblastic function. Indeed, Von Kossa and Alizarin Red staining also showed an apparent reduction in bone formation potential by miR-221-3p versus scrambled control (Figure 4).

Next, we established an in vitro co-culture model to research chondrocyte–osteoblast communication. As shown in (Figure 5a), chondrocytes were transfected with miR-221-3p mimic or scrambled control and seeded in the lower transwell chamber, as a molecular simulation of a mechanically challenged or resting cartilage layer. Then, untreated osteoblasts were seeded in the upper chamber, representing the subchondral bone osteoblasts.

After 48 h of coculture, the miR-221-3p expression in EVs isolated from the coculture medium was markedly increased by ~45 fold (*p* < 0.001) (Figure 5b). The qRT-PCR analysis also indicated that miR-221-3p was expressed in the scrambled control EVs with a ~26 ct value in 10 μg total extracellular RNA. This value increased to ~20 ct upon miR-221-3p mimic transfection of the chondrocytes (Appendix A). This result revealed that the cargo of EVs contained the signal of miR-221-3p and that the expression of miR-221-3p in EVs could be increased by transfection of the maternal cells. Meanwhile, miR-221-3p expression in osteoblasts was significantly increased in the miR-221-3p mimic group versus scrambled control (Figure 5c).

Further results demonstrated that direct target genes of miR-221-3p including CDKN1B/p27 (53%, *p* < 0.001), ARNT (34%, *p* < 0.001), TIMP-3 (51%, *p* < 0.001) and Tcf7l2/TCF4 (52%, *p* < 0.001) were significantly downregulated and osteogenic markers including COL1A1 and RUNX2 were inhibited by ~40% (*p* < 0.001) and 32% (*p* < 0.01) (Figure 6a–g).

### 2.4. Effect of Mir-221-3p Loaded EVs on Osteoblasts

To clarify, if the observed effect was indeed dependent on EV-transport of miR-221-3p from chondrocytes to osteoblasts, we isolated EVs from the supernatant of chondrocytes transfected with miR-221-3p mimic or scrambled control, then treated the osteoblasts with EVs directly.

Firstly, 4.0 × 10^4^ osteoblasts were treated with 5.0 × 10^8^ miR-221-3p-loaded EVs or scramble control-loaded EVs for 48 h. The number of chondrocytes used for EV isolation and the number of EV-treated osteoblasts was calculated with a ratio of 6:1, which was the same as in the coculture experiment. The results demonstrated a significant inhibitory effect of the same targets of miR-221-3p and osteogenic markers (Figure 7a–g). Briefly, the targets including CDKN1B/p27, ARNT, TIMP-3, and Tcf7l2/TCF4 were inhibited by ~33% (*p* < 0.001), 30% (*p* < 0.001), 32% (*p* < 0.001) and 37% (*p* < 0.001), and osteogenic markers including COL1A1 and RUNX2 were inhibited by ~33% (*p* < 0.001) and 21% (*p* < 0.01), respectively.

Thereafter, osteoblasts were treated with miR-221-3p-loaded EVs or scramble control EVs in osteogenic differentiation medium (ODM) for 2 weeks according to the same protocol as above. The cells demonstrated obvious suppressed capacity for mineralized nodule formation verified by Alizarin Red staining (Figure 8a,b) and quantification (Figure 8c). After osteoblasts treatment with miR-221-3p-loaded EVs in ODM for 2 weeks, the mRNA expression of the osteoblastic markers COL1A1, RUNX2, and OCN was measured by qRT-PCR (Figure 8d–f). The results showed a significant reduction in OCN expression (~86% *p* < 0.001), while no significant difference was observed in COL1A1 and RUNX2. This indicated an inhibition of the osteogenic capacity of osteoblasts by the miR-221-3p-loaded EVs.

Collectively, our results revealed that miR-221-3p can be transferred from chondrocytes to osteoblasts via EVs and that miR-221-3p-loaded EVs derived from chondrocytes can inhibit osteoblastic function in vitro. However, this inhibitory effect may also be mediated by another source: EVs’ parental cells modified by the transfection may also secrete EVs that are different not only by their miRNAs’ content, but also by the hundreds of miRNAs’ targets, which can also be modified during the transfection of the parental cells and the biogenesis of EVs. Furthermore, the presence of the miRNA of interest on its own does not strictly mean that the effect is mediated by the miRNA as one of the only demonstrations would be to remove the miRNA from the miRNA-modified EVs after its production and to see the abrogation of the effect.

## 3. Discussion

EVs are secreted by various cells that can exhibit a protective or destructive effect on target tissues by transferring their cargo in diverse physiological processes. It has been frequently reported that EVs can modulate cartilage degeneration and regeneration [25,26,27,28]. However, these studies have exclusively focused on the role of EVs on cartilage, despite the significance of the adjacent subchondral bone in the context of osteoarthritis [24]. Experimental evidence showed the mechanosensitive nature of miR-221 [16,29,30] and a paracrine effect from chondrocyte to chondrocyte [8] acting mainly on proliferation inside the same tissue. In this study, we demonstrate that chondrocyte derived EVs can inhibit the function of a completely different cell type (osteoblast) in a different tissue (bone), thereby establishing trans-tissue communication.

To date, multiple protocols have been compared for EVs isolation. Differential ultracentrifugation (UC) is acknowledged as the “gold standard”, despite the cumbersome centrifugal process and the limitation of mass production [31]. In the present study, the size of vesicles isolated by UC was found to be in a range of 30–200 nm (Figure 1a–d) and the characteristic markers of EVs (Alix, TSG101 and CD81) were identified by Western blot analysis (Figure 1f). This is consistent with current reports, which describe EVs as up to 200 nm in size vesicles with the above mentioned marker proteins [32,33,34,35]. Furthermore, TEM experiments (Figure 1c) confirmed typical morphological EV characteristics, which were consistent with the MISEV2018 guidelines [36].

EV transfer of miRNA has been extensively researched during the past decade. In particular, MSCs isolated from diverse sources (bone marrow, synovial membrane, adipose tissue, umbilical cord and embryonic cells) were reported to be able to transduce miRNA effects through secretion of EVs [37,38,39], mainly through leading to immunomodulation. Here, we demonstrated that chondrocytes also secrete miRNA through EVs and that the content of the EVs can be controlled by the expression of the miRNA. Similarly, Wang et al. reported that EVs containing miR-221-3p might attenuate OA by enhancing proliferation and migration of cartilage chondrocytes, but the exact mechanism was not further explored [27]. Considering the substantial mechanical/biochemical crosstalk between articular cartilage and subchondral bone [21,23,24], here, we studied EVs as a possible means for cell–cell communication between these tissues. In this regard, miR-221-3p was particularly interesting due to its mechanosensitive nature [16].

Generally, miRNAs can participate in different regulation mechanisms by inhibiting diverse targets, and our results partially confirmed that miRNA-221-3p targets CDKN1B/p27, TIMP-3, Tcf7l2/TCF4, and ARNT (Figure 2), which are all important cell cycle regulators. In brief, CDKN1B/p27 has been reported to be a versatile regulator of cell proliferation [40]; TIMP-3 can affect bone remodeling as an inhibitor of the matrix metalloproteinases [41]; Tcf7l2/TCF4 is an important component of the Wnt signaling pathway [42]; and ARNT—also known as a hypoxia-inducible factor (HIF)-1β—was reported to modulate the hypoxia-inducible factor (HIF) pathway [43]. In particular, targets of miR-221-3p were reported to participate in different biological processes and signaling pathways in the cancer field [44], e.g., in the Wnt signaling pathway, which has been also reported to participate in bone management by promoting osteoblast-relevant bone formation and inhibiting osteoclast-relevant bone resorption [45,46]. We found that the transfection of osteoblasts with miR-221-3p mimic led to significant overexpression of miR-221-3p, which consequently inhibited the osteogenic capacity of osteoblasts. This is consistent with another report that showed miR-221 overexpression could significantly decrease the mRNA expression levels of key osteoblast markers (after 24 h of treatment) in C2C12 cells by targeting RUNX2 [47]. Nevertheless, the long-term outcome of this inhibitory effect has never been shown.

Preceding results showed EVs secreted by chondrocytes contained miR-221-3p, the expression level was consistent with the maternal cells [48] and it changed with mechanical loading [16,30]. To simulate the interface between cartilage and bone, and to evaluate how the miR-221-3p secreted by chondrocytes can potentially affect the bone formation capacity of osteoblasts, we cocultured these two cells in an in vitro model and found that the signal of overexpressed miR-221-3p in chondrocytes was transferred to osteoblasts with inhibited osteogenic markers expression of COL1A1 and RUNX2 (Figure 6a–b), while the OCN (Figure 6c), a late marker for bone formation [49], was not inhibited. Since direct transfection of osteoblasts with miRNA-221-3p also decreased the OCN expression (Figure 3h), the unchanged expression of the late marker OCN after 2 weeks may be due to slower kinetics of the coculture.

To further clarify the effect of miRNA-221-3p-loaded EVs on osteoblasts, we illustrated that isolated EVs derived from the chondrocytes pre-transfected with miR-221-3p mimic could inhibit the bone formation capacity of osteoblasts both after a short time point of 48 h (Figure 7), as well as a two-week treatment (Figure 8). After 48 h treatment with isolated miRNA221-3p-loaded EVs, all miR-221-3p targets were reduced except for the late osteogenic marker OCN. After the longer treatment time of two weeks, the late osteogenic marker OCN was also significantly inhibited, while the expression of early osteogenic markers COL1A1 and RUNX2 returned to control values (Figure 8). This is consistent with Col1A1 and RUNX2, instead being early osteogenic differentiation markers [50,51]. Alizarin Red staining supported the inhibition of osteogenic differentiation by miR221-loaded chondrogenic EVs.

Based on our results, we hypothesize that the bone formation capacity of osteoblasts is regulated by the signal of miR-221-3p, mediated by EVs derived from chondrocytes. In previous studies, our group has demonstrated the existence of microchannel structures at the interface of bone and cartilage, which can potentially be pathways for the transportation of EVs and chondrocyte–osteoblast communication [22,23].

Figure 9 provides a schematic representation of how chondrocyte-secreted EVs that contain miR-221-3p may affect osteoblasts via the subchondral bone microchannel network. This could have great implications in understanding the bone–cartilage interface, since miRNAs mediated by EVs are known to be involved in the regulation of knee joint homeostasis and pathogenesis, while several mechanosensitive miRNAs have been demonstrated to be associated with the structural remodeling of bone and cartilage [16,52]. As miRNAs are protected by the EVs through the lipid bilayer membrane, longer communication routes from the maternal cell to the recipient cell would be conceivable [53] 

The early stages of OA are often characterized by bone loss caused by increased bone remodeling, followed by sclerosis of subchondral bone, osteophyte formation, and cartilage damage in end-stage OA [54]. This dynamic change in subchondral bone with the development of OA would be particularly interesting when considering the epigenetic regulation of miRNAs during the process. According to Hecht et al., the expression of miR-221 under mechanical loading was elevated only in healthy chondrocytes, not in OA chondrocytes [16]. Other studies have reported decreased expression of miR-221-3p in OA chondrocytes, which is consistent with our results regarding the IL-1β-treated chondrocytes (Appendix A). In light of our results, in both cases, a physiological inhibitory signal on the bone would be lost or diminished, which could eventually support the development of subchondral sclerosis. Hence, the present study may allow us to explain how the dynamic expression of miR-221-3p in cartilage under mechanic loading could affect the structure modification of subchondral bone.

Another physiological process in the course of OA development and subchondral bone remodeling is angiogenesis. It is suggested that angiogenesis may provide additional channels for cell communication and signaling, which includes EVs [55,56,57]. In accordance with this notion, we found that vascular endothelial growth factor (VEGF) was upregulated in both chondrocytes and osteoblasts after transfection with a miR-221-3p mimic (Appendix A). Additionally, some reports propose that downstream target TIMP-3 is not only involved in the regulation of the matrix metalloproteinases but also is related to angiogenesis during the invasion and metastasis of cancer cells and the protection of myocardial infarction [58,59]. Recently, Zhao et al. reported the TIMP3/TGF-β1 axis may be responsible for the deterioration and angiogenesis of chondrocytes under mechanical loading [60], which may be related to the regulatory role of miR-221-3p as well in this context.

An interesting observation was the non-linear effect of miR-221-3p enrichment on the inhibition of targets., i.e., 2725-fold overexpression of miR-221-3p led to a 50% reduction in target values compared to a 30% reduction in targets when a 2-fold miRNA enrichment was observed.

A possible explanation for this can be the sustained action by the miR-221-3p-loaded EVs over 48 h from the transfected chondrocytes. This might also be indirect evidence for a protective function of the lipid membrane on the miRNA degradation, compared to miRNA alone in cell–cell communication [53]. Zheng et al. have reported that a 3-fold increase in miR-221 in chondrocytes could significantly inhibit its target [13], which is in line with our observation. Murray et al. observed a significant overexpression (~25,000 fold) of miR-200b via miRNA200b mimics, while the target (Oxr1) expression was reduced by ~50% [61]. Furthermore, Genz et al. could show a 400-fold overexpression of miR-25 after miRNA transfection, leading to a 20–30% knockdown of the target genes FKBP14 and Adam-17 [62]. These observations confirm our results and the effects are in a similar range.

Furthermore, two studies have reported the miR-221 expression in articular cartilage under mechanical loading in vitro and in vivo, and the results demonstrated that the dynamic change in miR-221 expression in the articular cartilage was around 2-fold [16,30]. Considering the close proximity of bone and cartilage, it may be reasonable that the dynamic change in miR-221 expression in the osteoblasts induced by the transduction of miRNAs from chondrocytes is in the same vicinity. Likewise, it is well known that the regulatory effects of EV-mediated miRNAs are different from miRNAs alone in OA treatment [63]. If we extrapolate these findings, mechanosensitive miR-221-3p transported between cartilage and bone by EVs might also play a role in other physiological and pathophysiological processes at the border of cartilage and bone including fracture healing, pseudarthrosis formation, osteophyte development, and bone growth. Nevertheless, despite decades of research, our understanding of the exact mechanism of EVs’ biogenesis and secretion, transportation, and uptake is still in its infancy. Thus, further studies on the role of EVs-mediated miRNA regulation in the intercellular interaction are required to take advantage of this nanotechnology.

We must indicate that there are several limitations to this study. Firstly, the results were only generated in an in vitro model, which can only, in part, reflect the environment in vivo. If and how EVs are transferred from chondrocytes to osteoblasts in vivo and the mechanisms of how miR-221-3p transferred by EVs inhibits the bone formation capacity of osteoblasts need further research. Our results demonstrated that miR-221-3p-loaded EVs derived from chondrocytes can inhibit osteoblastic function in vitro. However, additional careful evaluations of this inhibitory effect need to be conducted. In particular, it should be noted that overexpression of miR-221-3p in the parental cells probably leads to additional effects due to activation of several miR targets in these parental cells, which may also result in other changes in the produced EVs. Consequently, the increased frequency of the miRNA does not necessarily guarantee that the observed effect is mediated only by the miRNA. Future approaches should therefore include the deletion of the miR-221 to see the possible abrogation of the effect. The effect of mechanics on miR-221-3p expression was stimulated by mimicking oligonucleotides, which cannot fully simulate the physiological and pathological expression of miRNA. Thus, the magnitude of the effect may be different in vivo. Last but not least, how to isolate truly purified EVs and set the physiologic amount of EVs in animal experiments and further in clinical trials would be a crucial issue to be resolved in future studies [64]. Therefore, additional in vitro and in vivo studies are planned to further investigate the cell–cell communication between chondrocytes and osteoblasts via extracellular vesicles.

## 4. Materials and Methods

### 4.1. Cell Isolation and Identification

Chondrocytes and osteoblasts were, respectively, isolated from the knee joint cartilage and calvarium of 3-day-old Wistar rats (Project number T20.1, Central Animal Experimental Facility, University Medical Center Göttingen, Germany) according to a modified enzyme digestion method [65]. Briefly, the snipped cartilage or calvarium was rinsed in pre-cooled PBS (PAN-Biotech, Aidenbach, Germany), digested for 30 min in 0.25% trypsin (PAN-Biotech, Aidenbach, Germany) in a 37 °C water bath, followed by digestion in 0.1% collagenase II (Sigma Aldrich, St. Louis, MO, USA) at 37 °C for 4 h. Subsequently, cell suspension was centrifuged and the resulting pellet was resuspended with DMEM-low glucose (Thermo Fisher Scientific, MA, USA) containing 10% fetal bovine serum (FBS) and 1% penicillin/streptomycin antibiotics (PAN-Biotech, Aidenbach, Germany) at 37 °C with 5% CO_2_. The medium and the passage cells were changed twice a week and in the case of 90% confluency, respectively.

The characteristics of the chondrocytes were identified by immunofluorescence. In brief, sections were firstly incubated overnight with primary collagen II antibody (Abcam, Cambridge, UK, 1:50) at 4 °C, and then, incubated with the HRP-coupled secondary antibody (Abcam, Invitrogen, USA, 1:10.000) for one hour at room temperature. Finally, the sections were visualized under a fluorescence microscope (Leica DMi8, Wetzlar, Germany). To identify the osteoblasts, they were seeded in 24-well plates at a density of 5 × 10^4^ and cultured in osteogenic differentiation medium (ODM, DMEM-Low glucose containing 10% FCS, 1% penicillin-streptomycin, 0.2 mM ascorbic acid-2-phosphate and 10 mM β-glycerophosphate) for 2 weeks, followed by Von Kossa as well as Alizarin Red staining (Appendix A). Meanwhile, the osteoblast markers COL1A1, RUNX2, and OCN and the expression of miR-221-3p in osteoblasts were tested by qRT-PCR (Appendix A).

### 4.2. Establishment of OA Model In Vitro

To establish an in vitro OA model, IL-1β (Peprotech, Rocky Hill, NJ, USA) was adopted to treat chondrocytes as described previously [65]. Briefly, 5 × 10^4^ chondrocytes were seeded in 24-well plates and treated with 10 ng/mL IL-1β in the complete medium. After 24 h, the cells were harvested for further analysis.

### 4.3. Transfection

MiR-221-3p mimicking oligonucleotides and scrambled control oligonucleotides (QIAGEN, Hilden, Germany) were transfected into chondrocytes at a ratio of 50 nM/5 × 10^4^ cells, using Lipofectamine RNAiMAX Transfection Reagent (Thermo Fisher Scientific, MA, USA) according to the manufacturer’s instructions. After 6 h of transfection, the transfected chondrocytes were washed three times with PBS and changed with medium without containing FBS. After 48 h, the cells were collected for further analysis, while the supernatant was collected to isolate EVs loaded with miR-221-3p or scramble control (see Section 4.5.). Similarly, osteoblasts were transfected with the same protocol every three days and 4 times in total. The transfected osteoblasts were collected for further analysis after two weeks.

### 4.4. Coculture of Chondrocytes and Osteoblasts

To establish a cell–cell communication model in vitro, 5 × 10^4^ osteoblasts were seeded in a transwell with a pore size of 3 µm (Merck KGaA, Darmstadt, Germany), and cocultured with 3 × 10^5^ chondrocytes pre-transfected with miR-221-3p mimic and scrambled control in 6-well plates for 48 h. A complete medium containing 10% FBS free of EVs (ultracentrifuged for 16 h) and 1% P/S (2.6 mL for chondrocytes and 1.5 mL for osteoblasts) was used for the process. The cells and supernatant were harvested for further analysis.

### 4.5. EVs Isolation and Identification

According to the recommendation from ISEV (International Society for Extracellular Vesicles), “extracellular vesicle” (EV) was adopted to name the vesicles in the present study [38]. EVs were isolated from a conditioned supernatant of chondrocytes with the classical ultracentrifuge method [66,67]. In brief, chondrocytes were cultured in a complete medium until 80% confluence. After a PBS wash, cells were cultured in a DMEM-low glucose medium without additives for 48 h. Afterwards, the conditioned medium was collected and centrifuged at 500× *g* for 5 min, and at 2000× *g* for 20 min, to remove dead cells and clear debris and large vesicles, respectively. Thereafter, 100 mL supernatant underwent ultracentrifugation in polyallomer tubes (38.5 mL, Beckman Coulter, Brea) for 35 min at 36,000× *g*, followed by 2 h at 110,000× *g* (Optima XPN-80 Ultracentrifuge, SW32 Ti rotor, Beckman Coulter, Brea). Finally, the EV pellets were dissolved in 100 μL PBS and stored at 4 °C for further experiments.

The NanoSight platform (NanoSight LM10, Malvern 24alytical, Kassel, Germany) was used to measure the size distribution and particle concentration of EVs. A LEO912 transmission electron microscope (Carl Zeiss Microscopy, Oberkochen, Germany) and an On Axis 2k CCD camera (TRS-STAR, Stutensee, Germany) were used to observe the morphology of EVs. The specific markers of EVs were detected by Western blot analysis (Section 4.7.)

### 4.6. Target-Gene Prediction and Data Analysis

Public online websites including Targetscan (Retrieved 31 December 2020, from http://www.targetscan.org/vert_72/), miRWalk (Retrieved 31 December 2020, from http://zmf.umm.uni-heidelberg.de/apps/zmf/mirwalk2/), and miRDB (Retrieved 31 December 2020, from http://mirdb.org/) were used to predict the target gene of miR-221-3p. Afterwards, a Venn diagram (Retrieved 31 December 2020, from http://bioinformatics.psb.ugent.be/webtools/Venn/) was used to compile all the predicted targets. To quantify the calcium deposition of osteoblasts after Alizarin red staining, the publicly available software Image J (Version: 1.53m) was adopted (threshold = 120).

### 4.7. Western Blot Analysis

Total protein was extracted from chondrocyte samples and EV samples using RIPA buffer (25 mM Tris-HCl pH 7.6,150 mM NaCl, 1% NP-40, 1% Natrium-Deoxycholate, 0,1% SDS). After washing with precooled PBS, protein concentration was measured with a BCA Protein Assay Kit (Thermo Fisher Scientific, MA, USA). Protein samples were mixed with Laemmli (Bio-Rad, US) with a ratio of 4:1, and then heated for 5 min at 95 °C. Equal amounts of protein were resolved on 12% SDS-PAGE gels and transferred to polyvinylidene difluoride (PVDF) membranes (Bio-Rad, CA, USA). Membranes were blocked with 5% skimmed milk for 1 h and incubated with the primary antibodies COX-2, β-actin, CD81, Alix, and Tsg101 overnight at 4 °C (Appendix A). After washing with TBST buffer, PVDF membranes were incubated with HRP coupled secondary antibody (Thermo Fisher, MA, USA, 1:10.000) for 1 h. Subsequently, the rinsed membranes were soaked in ECL chemiluminescence reagent (Bio-Rad, CA, USA), and the blots were exposed with the imaging system ChemiDoc XRS + (Bio-Rad, CA, USA). The results were further analyzed via Image Lab Software (Bio-Rad, CA, USA).

### 4.8. RNA Isolation and Real-Time qRT-PCR

Total RNA was extracted using Trizol reagent (Invitrogen, Waltham, MA, USA) according to the manufacturer’s instructions. RNA concentration and quality were detected with a DS-11 FX + integrated spectrophotometer (Thermo Fisher Scientific, MA, USA). For quantification analysis of mRNA, 1000 ng RNA was reversely transcribed and amplified, while the gene expression was relative to GAPDH. For miRNA analysis, 10 ng RNA was reversely transcribed and amplified using the miRCURY LNA miRNA Kit (QIAGEN, Hilden, Germany) according to the protocols of the manufacturer, while miRNA expression was relative to U6 small nuclear RNA (snRNA). All the calculations for the relative results adopted the standard 2−ΔΔCt method.

### 4.9. Software and Statistical Analysis

All data were statistically analyzed using GraphPad Prism 5 Software (GraphPad, CA, USA). The Mann–Whitney U-test was used to assess the difference between two groups. Unless otherwise stated, results were shown as mean values with standard deviations, and statistical differences were considered significant when the *p* value was <0.05. All the experiments were repeated with three biological replicates and three technical replicates each. Figures were partly with BioRender.com (May 2021).

## 5. Conclusions

In conclusion, the present study herein demonstrated that EVs from chondrocytes can transfer the mechanosensitive miR-221-3p to osteoblasts, acting as an intercellular messenger and reducing osteoblastic bone formation in vitro. This facilitates a novel perspective on how soft tissues can transduce mechanical cues to adjacent harder tissues through EVs as molecular messengers.

## Figures and Tables

**Figure 1 ijms-22-13282-f001:**
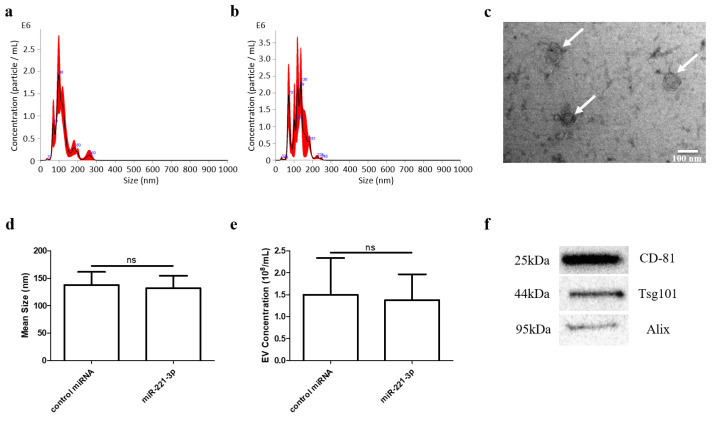
EVs were isolated by ultracentrifugation and identified by NTA, TEM and Western blotting. Particle size distribution of scramble miRNA-loaded (**a**) and miRNA-221-3p-loaded EVs (**b**) isolated by ultracentrifugation was determined by nanoparticle tracking analysis (NTA). Typical morphology (white arrows) of EVs was observed under TEM (**c**) scale bar 100 nm. Transfection of miRNA-221-3p did not affect average size (**d**) and concentration (**e**) of EVs compared to control RNA transfections. Exosome-specific protein markers CD81, TSG101 and Alix were detected by Western blotting (**f**). ns stands for not statistically significant.

**Figure 2 ijms-22-13282-f002:**
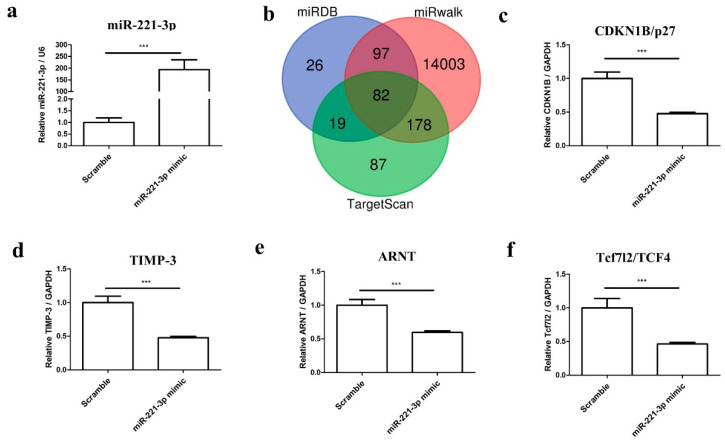
MiR-221-3p was increased by mimic transfection and the targets were predicted and identified in chondrocytes. (**a**) The transfection efficiency of the miR-221-3p mimic was confirmed by RT-qPCR relative to U6. (**b**) Predicted targets of miR-221-3p were identified using three independent platforms, i.e., miRDB, TargetScan, and miRwalk. (**c**–**f**) Chondrocytes were transfected with miR-scramble and miR-221-3p mimic, while the expression level of predicted targets was measured relative to GAPDH by RT-qPCR and *** *p* < 0.001 vs. corresponding control.

**Figure 3 ijms-22-13282-f003:**
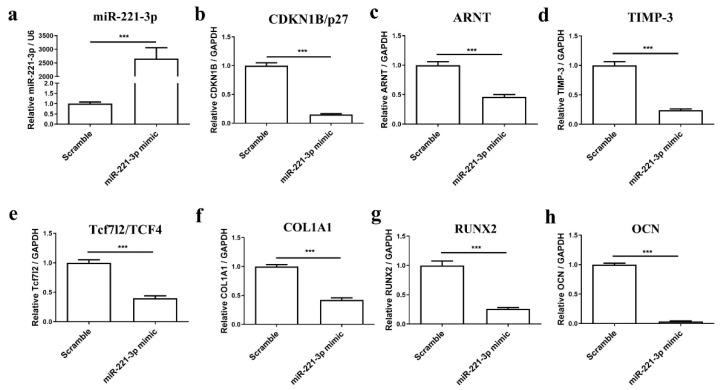
MiR-221-3p can act in osteoblasts. Osteoblasts were transfected with miR-scrambled or miR-221-3p mimic for 2 weeks in ODM, the expression level of miR-221-3p (**a**), the expression of putative targets of miR-221-3p, CDKN1B/p27 (**b**), ARNT (**c**), TIMP-3 (**d**), and Tcf7l2/TCF4 (**e**) and osteogenic markers COL1A1 (**f**), RUNX2 (**g**), OCN (**h**) were measured by RT-qPCR. CDKN1B/p27, ARNT, TIMP-3, Tcf7l2/TCF4, COL1A1, RUNX2, OCN relative to GAPDH, and miR-221-3p relative U6. *** *p* < 0.001 vs. corresponding control.

**Figure 4 ijms-22-13282-f004:**
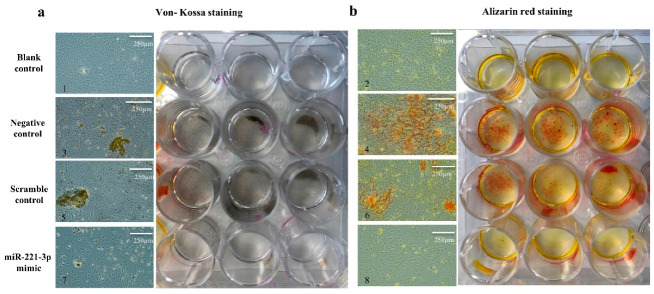
MiR-221-3p decreases osteoblast mineralization. The microscopic and macroscopic results of Von Kossa staining (**a**) and Alizarin Red staining (**b**) for osteoblasts after 2 weeks’ culture in different conditions. Negative control without miRNA treatment (3, 4), scramble control (5, 6) and miR-221-3p mimic (7, 8) were cultured with ODM. The undifferentiated blank control group (1, 2) was cultured with BM. BM = Basal medium, ODM = Osteogenic differentiation medium.

**Figure 5 ijms-22-13282-f005:**
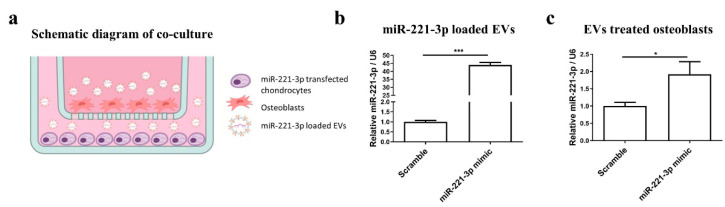
Chondrocyte signals through miR-221-3p can be transferred to osteoblasts via extracellular vesicles (**a**). Osteoblasts and chondrocytes were cocultured in a transwell system. miR-221-3p was overexpressed in EVs secreted by transfected chondrocytes (**b**). miR-221-3p in osteoblasts was significantly increased when cocultured with transfected chondrocytes (**c**). miRNA expression relative to U6. * *p* < 0.05 and *** *p* < 0.001 vs. corresponding control. This figure was partly created with BioRender.com (May 2021).

**Figure 6 ijms-22-13282-f006:**
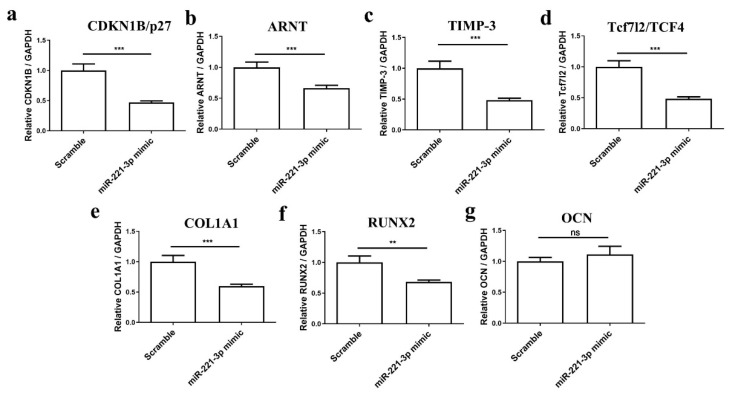
Secreted molecules from chondrocytes inhibited targets expression of miR-221-3p and osteogenic markers in osteoblasts. CDKN1B/p27 (**a**), ARNT (**b**), TIMP-3 (**c**), and Tcf7l2/TCF4 (**d**), and osteogenic markers (**e**–**g**), COL1A1, RUNX2, and OCN in osteoblasts cocultured with chondrocytes transfected with miR-221-3p mimic compared with scrambled control. Relative to GAPDH. ns stands for not statistically significant, ** *p* < 0.01, and *** *p* < 0.001 vs. corresponding control.

**Figure 7 ijms-22-13282-f007:**
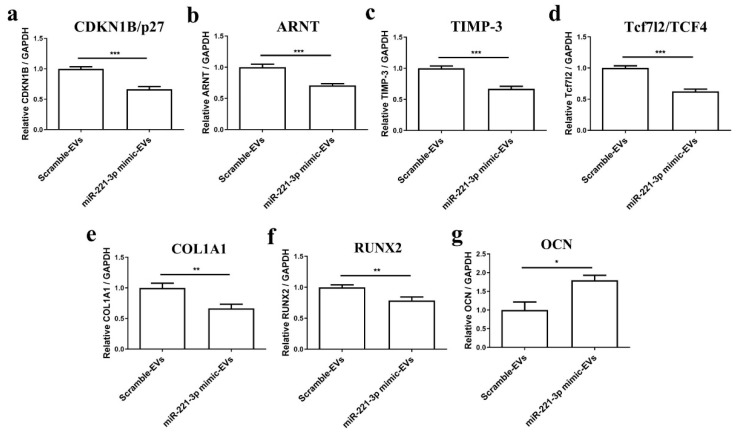
Chondrocytes secreted miR-221-3p-EVs inhibited osteogenic markers and targets expression of miR-221-3p in osteoblasts after 48 h coculture. The expression level of and targets expression (**a**–**d**) of miR-221-3p CDKN1B/p27, ARNT, TIMP-3, and Tcf7l2/TCF4, and osteogenic markers (**e**–**g**), COL1A1, RUNX2, and OCN in osteoblasts cocultured with EVs modified by mir-221-3p mimic compared with miRNA scramble after 48 h. Relative to GAPDH., * *p* < 0.05, ** *p* < 0.01 and *** *p* < 0.001 vs. corresponding control.

**Figure 8 ijms-22-13282-f008:**
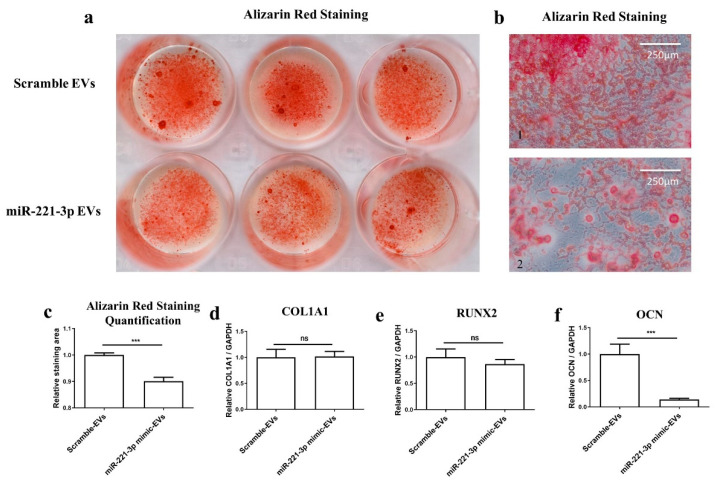
Osteoblasts were treated with EVs isolated from chondrocytes supernatant with or without miR-221-3p overexpression for 2 weeks in ODM. The microscopic (**a**) and macroscopic (**b**) result of Alizarin Red staining, respectively. Quantitative analysis of the Alizarin Red staining area through ImageJ (**c**), threshold = 120. The relative expression level of osteogenic markers (**d**–**f**), COL1A1, RUNX2, and OCN in the two groups. Relative to GAPDH. ns stands for not statistically significant, *** *p* < 0.001 vs. corresponding control. ODM = Osteogenic differentiation medium.

**Figure 9 ijms-22-13282-f009:**
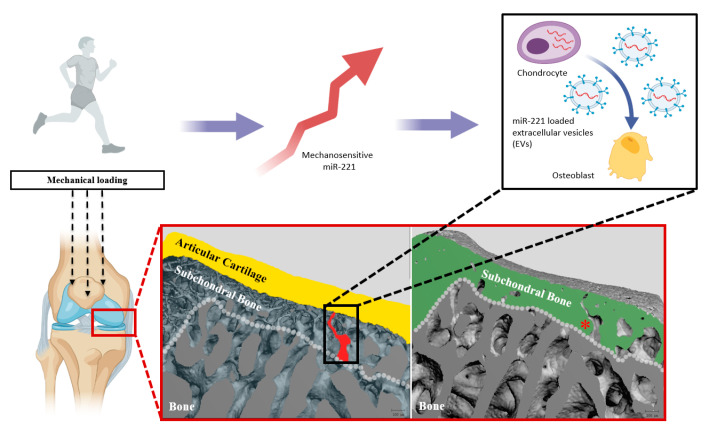
Theoretical scheme of how chondrocyte-secreted EVs harboring miR-221-3p may affect osteoblasts in the physiological condition of joint. The healthy knee joint can be affected by mechanical loading, which increases miR-221-3p expression in the cartilage chondrocytes. Chondrocytes secrete EVs that contain these miRNAs, which can then be taken up by osteoblasts via the microchannel network of the subchondral bone. Subchondral bone (green) and its intricate microarchitecture (red asterisk) are illustrated by the 3D-reconstructed images acquired by high-resolution micro-computed tomography. In the inverted model, the articular cartilage is demonstrated in yellow, while the continuous nature of porous structures (i.e., microchannels; one of which is marked in red color) is visible throughout the entire subchondral bone. These microchannels may be the main pathways of bone–cartilage communication.

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
