# Peer review of "Extracellular Vesicles Allow Epigenetic Mechanotransduction between Chondrocytes and Osteoblasts"

_ijms, 2021, doi:10.3390/ijms222413282_

Round 1

Reviewer 1 Report

Thank you for the possibility to review this interesting article. The main question in this article is a novel perspective on a possible communication pathway of a mechanically induced epigenetic signal through EVs. This is very interesting topic according to future OA therapy possibilities or the novel drugs development. Paper is well written with the clear figures. Text is clear and easy to read even for the beginning scientist. The study question is answered. I would like to ask you to add more information about how your data can be used in the future research planning. Please consider to use additional person to correct typographic errors and repeating references (32 and 44).

Author Response

We would like to thank all the reviewers for their careful review and valuable comments. According to these suggestions, we now have revised the manuscript and provided a point-by-point response to each raised comment below. In the manuscript, the revised portions are marked in light yellow.

Reviewer 2 Report

Dear authors,

You will find in the PDF attached all detailed comments. This work is interesting, concise, clearly explained and well written, but I think there are two major things to improve in the manuscript:

1) some claims are not sufficiently supported in my opinion, in particular : 

- If I am correct, a 2500 fold enrichement in miRNA led to a about 50% reduction in targets versus about 30% with a 2 fold enrichment. Although I do not expect a linear relation, this is very far from being linear. It is in my opinion necessary to see whether this "incredible" effect may be mediated by miRNAs at such a low dose by making a simple experiment : 

a) deciphering what is the dose of lipofectamine needed to achieve an about 2 fold increase in miRNA in osteoblasts

b) look at the expression level of all the genes of interest described in figure 6 and see whether such an efficient reduction of about 30% is visible.

If yes, that would add a great strenght to the demonstration 
If not, it would mean that the incredible effect demonstrated may be mediatd by something else... (e.g. EVs that are coming from a cell that was modified by the expression of the miR of interest, instead of the miR on its own)

2) some technical details must be detailed, in particular : 

  • it has to be explained in the main text that cells were transfected and not EVs, and also the transfection protocol (was there washing steps between lipofectamine and EV production) ?  If not, EV preparation is contaminated by lipofectamine/RNA aggregates and the effect claimed is probably coming from lipofeactamine instead of EVs.
  • Please indicate number of biological and technical replicates
  • There seems to be no quantification at all of the amount of EVs used on osteoblasts. It is necessary to provide the following informations in the manuscript : 
    1) How many EVs per recipient cell were used in these experiments
    2) what is the amount of EVs produced per cell in culture 
    3) based on these two numbers, to clearly say if it is the case that the number of EVs put in the experiment was supra physiologic (if EV per cell received per cell > to the amount of EV produced by osteoblasts), and quantify how many fold. 

    If it appeared to be highly supraphysiologic, it would of course be difficult to attribute the previosly described effects observed to EVs. 

Once these issues will be clarified and experiments added or explained why unnecessary, reviewers may be able to assess whether the final claim is well supported

Author Response

(The authors gave the same response as above.)

Round 2

Reviewer 2 Report

Dear authors, 

thank you for your detailed response. As you chose not to launch new experiments (a perfectly understandable solution) and that you discussed the limits of the experiments, i feel that the manuscript is better than the previous version. 

In my opinion,  only one point left needs to be clarified and discussed in the manuscript : 

You claim that : " Collectively, our results revealed that miR-221-3p can be transferred from chondrocytes to osteoblasts via EVs and that miR-221-3p loaded EVs derived from chondrocytes can inhibit osteoblastic function in vitro."

I would add and discuss : 

1) the fact that this effect may also be mediated by another effect : EV parental cells modified by the transfection may also express EVs that are different not only by their miR content, but also by the hundred of miR targets that will be modified by the parental cell and its EVs. 

The presence of the miRNA of interest on its own does not strictly mean that the effect is mediated by the miRNA as one of the only demonstration would be to remove the miRNA from the miRNA-modified EVs after its production and to see the abrogation of the effect. It would therefore be interesting to discuss this in the discussion and be a little bit less affirmative on the mechanism of action (EVs mediate the effect for sure, maybe/probably via the miRNA inside).  

One may argue that the downregulation of the miR targets is sufficient to affirm the miR mediated effect, but I guess such "limited" decrease in expression may also be found in the presnece of the phenotype you observe (inhibition of osteocyte function), whatever is its trigger (an EV mediated receptor ligand interaction, miRNA transfer, etc...). 

2) that this will be tested in vivo

Thank you for this interesting discussion, your argumentation was very well constructed (although I would have prefered data !)

Best regards

Author Response

Reviewer 2:

Dear authors,

thank you for your detailed response. As you chose not to launch new experiments (a perfectly understandable solution) and that you discussed the limits of the experiments, i feel that the manuscript is better than the previous version.

Response: The authors would like to thank the referee for the positive evaluation of our manuscript. We are pleased to hear that the manuscript quality is increased compared to the first version.

In my opinion,  only one point left needs to be clarified and discussed in the manuscript :

You claim that : " Collectively, our results revealed that miR-221-3p can be transferred from chondrocytes to osteoblasts via EVs and that miR-221-3p loaded EVs derived from chondrocytes can inhibit osteoblastic function in vitro."

I would add and discuss :

1) the fact that this effect may also be mediated by another effect : EV parental cells modified by the transfection may also express EVs that are different not only by their miR content, but also by the hundred of miR targets that will be modified by the parental cell and its EVs.

The presence of the miRNA of interest on its own does not strictly mean that the effect is mediated by the miRNA as one of the only demonstration would be to remove the miRNA from the miRNA-modified EVs after its production and to see the abrogation of the effect. It would therefore be interesting to discuss this in the discussion and be a little bit less affirmative on the mechanism of action (EVs mediate the effect for sure, maybe/probably via the miRNA inside). 

One may argue that the downregulation of the miR targets is sufficient to affirm the miR mediated effect, but I guess such "limited" decrease in expression may also be found in the presnece of the phenotype you observe (inhibition of osteocyte function), whatever is its trigger (an EV mediated receptor ligand interaction, miRNA transfer, etc...).

2) that this will be tested in vivo

Response: Thank you very much for your positive comments and constructive suggestions. We have now added an extra paragraph according to your comments in the revised manuscript and highlighted the new section in yellow (Page 11/Line 365-373 + 378-380).

Thank you for this interesting discussion, your argumentation was very well constructed (although I would have prefered data !)

Best regards

We want to thank the reviewer again for his/her time and careful evaluation of the manuscript. Thanks to these comments, we hope that we were able to further improve the quality of the manuscript.